# Connection between the Gut Microbiome, Systemic Inflammation, Gut Permeability and FOXP3 Expression in Patients with Primary Sjögren’s Syndrome

**DOI:** 10.3390/ijms21228733

**Published:** 2020-11-19

**Authors:** Antonio Cano-Ortiz, Aurora Laborda-Illanes, Isaac Plaza-Andrades, Alberto Membrillo del Pozo, Alberto Villarrubia Cuadrado, Marina Rodríguez Calvo de Mora, Isabel Leiva-Gea, Lidia Sanchez-Alcoholado, María Isabel Queipo-Ortuño

**Affiliations:** 1Servicio de Oftalmología, Hospital de la Arruzafa. Universidad de Córdoba, 14012 Córdoba, Spain; antoniocanoortiz@gmail.com (A.C.-O.); alberto.membrillo@gmail.com (A.M.d.P.); alvillarrubia@yahoo.com (A.V.C.); 2Unidad de Gestión Clínica Intercentros de Oncología Médica, Hospitales Universitarios Regional y Virgen de la Victoria, Instituto de Investigación Biomédica de Málaga (IBIMA)-CIMES-UMA, 29010 Málaga, Spain; auroralabordaillanes@gmail.com (A.L.-I.); isaacplazaandrade@gmail.com (I.P.-A.); 3Facultad de Medicina, Universidad de Málaga, 29071 Málaga, Spain; 4Servicio de Oftalmología, Hospital Regional Universitario, 29009 Málaga, Spain; marocalmo@gmail.com; 5Unidad de Endocrinología Pediátrica, Hospital Materno-Infantil, 29009 Málaga, Spain; isabeleiva@hotmail.com

**Keywords:** primary Sjögren’s syndrome, gut microbiota, inflammation, intestinal permeability, FOXP3 expression

## Abstract

The aims of this study were to explore intestinal microbial composition and functionality in primary Sjögren’s syndrome (pSS) and to relate these findings to inflammation, permeability and the transcription factor Forkhead box protein P3 (FOXP3) gene expression in peripheral blood. The study included 19 pSS patients and 19 healthy controls matched for age, sex, and body mass index. Fecal bacterial DNA was extracted and analyzed by 16S rRNA sequencing using an Ion S5 platform followed by a bioinformatics analysis using Quantitative Insights into Microbial Ecology (QIIME II) and Phylogenetic Investigation of Communities by Reconstruction of Unobserved States (PICRUSt). Our data suggest that the gut microbiota of pSS patients differs at both the taxonomic and functional levels with respect to healthy controls. The gut microbiota profile of our pSS patients was characterized by a lower diversity and richness and with Bacteroidetes dominating at the phylum level. The pSS patients had less beneficial or commensal butyrate-producing bacteria and a higher proportion of opportunistic pathogens with proinflammatory activity, which may impair intestinal barrier function and therefore contribute to inflammatory processes associated with pSS by increasing the production of proinflammatory cytokines and decreasing the release of the anti-inflammatory cytokine IL-10 and the peripheral FOXP3 mRNA expression, implicated in the development and function of regulatory T cells (Treg) cells. Further studies are needed to better understand the real impact of dysbiosis on the course of pSS and to conceive preventive or therapeutic strategies to counteract microbiome-driven inflammation.

## 1. Introduction

Primary Sjögren’s syndrome (pSS) is a systemic inflammatory autoimmune disease characterized by chronic inflammation of the exocrine glands, in particular the salivary and lacrimal glands, resulting in oral and ocular dryness (sicca) complaints [1]. Additionally, patients develop fatigue, depression, and inflammation in extraglandular tissues including the brain, lungs, and gastrointestinal tract [2]. 

The etiology of pSS remains to be elucidated, although both environmental and genetic factors are believed to be involved in the pathogenesis [3]. Individuals with autoimmune diseases such as spondyloarthritis [4], rheumatoid arthritis [5] and lupus erythematosus [6] have been found to have gut microbiome alterations compared to healthy controls. Several recent studies have described differences in the gut and oral microbiota of pSS patients compared with healthy and symptom controls. The study by de Paiva et al. showed a significant increase in *Staphylococcus aureus* and *Candida albicans* accompanied by a decrease in *Leptotrichia* and *Fusobacterium* in the oral mucosa and the tongue samples of SS patients. In addition, high levels of *Lactobacillus* spp. were found in supragingival plaque samples [7]. Concerning gut microbiota, these authors observed depletion of *Faecalibacterium*, *Bacteroides*, *Parabacteroides* and *Prevotella* and enhancement of *Escherichia, Shigella*, and *Streptococcus* genera [7]. Mandl et al. found that severe intestinal dysbiosis was more prevalent in pSS patients in comparison to healthy controls and this dysbiosis was associated with clinical and laboratory markers of systemic disease activity and with gastrointestinal inflammation [8].

The gut microbiota is involved in maintaining balance in immune responses between regulatory T cells (Tregs) and T helper 17 (Th17) cells on the mucosal surface and acts as a trigger for the induction of autoimmunity, such as in rheumatic diseases [9].

The strong anti-inflammatory effects exhibited by Treg cells expressing the transcription factor Forkhead box protein P3 (FOXP3) are necessary to balance immune responses and prevent chronic inflammation. In fact, the depletion of these Treg cells has been reported to result in autoimmune diseases [10]. The interplay between the gut microbiota, the epithelium and the immune cells of the gastrointestinal mucosa has been shown to have significant effects on the local and systemic immune systems [11] and may both decrease and increase local and systemic inflammatory disease [12]. In pSS patients, it has been considered that, the integrity of the gastrointestinal epithelium and its barrier function may be altered through inflammation and diminished secretions of the exocrine glands, producing an altered microbiota–host immune system interaction [13,14]. Exocrine gland dysfunction is driven by high levels of key factors such as proinflammatory cytokines including interleukin (IL)-12, IL-6, interferon-gamma (IFN-gamma) and tumor necrosis factor (TNF)-alpha; serum autoantibodies including antinuclear antibodies (ANA), antibodies against anti-Ro/Sjögren’s syndrome autoantibody A (Ro/SSA) and anti La/Sjögren’s syndrome autoantibody B (La/SSB), and rheumatoid factor (RF); and significant infiltration of T and B cells [15]. Gut microbiome dysbiosis can modulate systemic inflammation but systemic inflammation can also reduce beneficial gut bacteria, promoting the growth of commensal bacteria with the potential to be pathogenic [16,17]. 

The aims of this study were therefore to explore intestinal microbial composition and functionality in pSS and to relate these findings to inflammation, permeability and FOXP3 gene expression in peripheral blood.

## 2. Results

### 2.1. Clinical Characteristics of the pSS and the Healthy Control Groups

Demographic and clinical characteristics of the participants are shown in Table 1. No significant differences in age, sex, body mass index (BMI), race, smoking, total cholesterol, triglycerides, high-density lipoprotein (HDL)-cholesterol, low-density lipoprotein (LDL)-cholesterol or fasting glucose were found between study groups (*p* > 0.05). Only the serum levels of c-reactive proteins (CRP) were significantly increased in pSS patients compared to healthy controls. Comorbidities in the pSS patients included type 2 diabetes (*n* = 1), hypothyroidism (no Hashimoto’s thyroiditis) (*n* = 3), fibromyalgia (*n* = 2), oral ulcers (*n* = 1) and arthralgia (*n* = 3) but none of the pSS patients had a comorbid autoimmune disease. We included patients that had not taken steroids or immunosuppressant for at least 3 months prior to sample collection.

Adherence to the Mediterranean diet assessed via a 14-item food frequency questionnaire reflected high adherence to the Mediterranean diet in both study groups.

### 2.2. Taxonomic Composition and Diversity of Gut Microbiota

A total of 2,473,675 reads of the 16S rDNA gene V2–V9 region were generated from the 38 analyzed samples, with an average of 65,096.71 (±10,825.11 standard deviation (SD)) reads for each sample, ranging from 630,489 to 2582. A total of 52,844.28 high quality reads were obtained after trimming and filtering. In the operational taxonomic unit (OTU) clustering process, a total of 6473 OTUs were obtained, and after alignment of the OTUs representative sequences, 2582 OTUs with a relative abundance higher than 1% in at least four samples (97% similarity cut-off) were identified. In the taxonomic assignment process, these OTUs were binned in 7 phyla, 39 families, 45 genera and 53 species using the Quantitative Insights into Microbial Ecology (QIIME2) pipeline, with Greengenes v13.8 as the reference database for sequence classification and alignment.

To compare the alpha diversity, the Chao1, ACE, Shannon and Simpson indices were calculated after rarefying the OTU table to 1731 sequences, the size of the smallest sample to obtain equal sequencing depth. These analyses revealed a significant decrease in richness (Chao1 and ACE) and diversity (Shannon and Simpson) in pSS patients compared to healthy controls (Chao1 *q* < 0.001, ACE *p* < 0.001, Shannon *p* = 0.0017 and Simpson *p* = 0.05) (Figure 1A,B).

Beta diversity was evaluated by the Bray-Curtis Index metric at the genus level. Significant shifts in community membership and structure were observed between the study groups in the principal coordinates analysis (PCoA) plots (*p* = 0.007, ANOSIM). As shown in Figure 2 a significant separation in clustering pattern between pSS and healthy controls could be observed from PC1 and PC2 scores that accounted for 10% and 12.7% of the total variations.

### 2.3. Comparative Analysis of the Gut Microbial Composition of Patients with pSS and Healthy Controls 

Compositional differences in gut microbiota at the level of phylum, family, genus and species were found between both study groups. A core microbiome consisting predominantly of Firmicutes, Bacteroidetes, Proteobacteria, WS6 and Actinobacteria phyla was found in fecal samples from both groups, accounting for up to 90% of sequences on average. The remaining bacterial population belonged to the other phyla (Tenericutes, Fusobacterium and Lentisphaerae), which had a relative abundance lower than 1% in both groups. At this phylum level, we found a significant rise in Bacteroidetes (40.24% pSS vs. 33.05% HC, *p* < 0.001, *q*-value < 0.001) and Proteobacteria (11.22% pSS vs. 6.98% HC, *p* = 0.0020, *q*-value = 0.017) in pSS patients compared to healthy controls. While Firmicutes (20.12% pSS vs. 31.22% HC, *p* = 0.0021, *q*-value = 0.02) and Actinobacteria (2.54% pSS vs. 6.12% HC, *p* = 0.012, *q*-value = 0.048) were significantly enriched in healthy controls compared to pSS patients (Figure 3).

Of the 39 families identified, 10 showed significant differences in their abundance between pSS patients and healthy controls. In the Firmicutes families, two families were significantly more frequent in pSS patients compared with healthy controls such as Clostridiaceae (6.87% pSS vs. 1.34% HC, *p* < 0.001, *q*-value < 0.001) and Veillonellaceae (3.75% pSS vs. 1.86% HC, p = 0.0032, *q*-value = 0.02) On the other hand, the frequency of Ruminococcaceae (7.65% pSS vs. 27.53% HC, *p* < 0.001, *q*-value = 0.002) and Lachnospiraceae (8.44% pSS vs. 20.10% HC, *p* < 0.001, *q*-value = 0.001) was significantly reduced in pSS compared with healthy controls. Within the Bacteroidetes, three different families were significantly higher in pSS patients when compared with healthy controls: Prevotellaceae (20.18% pSS vs. 2.05% HC, *p* < 0.001, *q*-value < 0.001), Rikenellaceae (14.03% pSS vs. 4.44% HC, *p* = 0.013, *q*-value = 0.05) and Odoribacteraceae (2.96% pSS vs. 1.60% HC, *p* = 0.009, *q*-value = 0.046). Bacteroidaceae (25.12% pSS vs. 29.79% HC, *p* = 0.011, *q*-value = 0.046) and Porphyromonadaceae (2.72% pSS vs. 2.57% HC, *p* = 0.0021, *q*-value = 0.017) were significantly lower in pSS patients than in healthy controls. In Actinobacteria, a significant decrease in Bifidobacteriaceae (2.45% pSS vs. 4.43% HC, *p* < 0.001, *q*-value < 0.001) was found in pSS patients compared with healthy controls. While in the Proteobacteria family, there was a significant increase in the abundance of Enterobacteriaceae (3.77% pSS vs. 1.50% HC, *p* < 0.001, *q*-value = 0.001) in pSS patients (Figure 4).

A total of 15 genera exhibited significant compositional changes in pSS patients compared to healthy controls. In the pSS patients, we found a significant increase in the abundance of *Prevotella* (21.43% pSS vs. 4.14% HC, *p* = 0.0017, *q*-value = 0.014), *Escherichia* (2.78% pSS vs. 1.05% HC *p* = 0.011, *q*-value = 0.046), *Clostridium* (2.48% pSS vs. 1.11% HC, *p* < 0.001, *q* value = 0.007), *Enterobacter* (5.34% pSS vs. 1.30% HC, *p* = 0.0034, *q*-value = 0.02), *Veillonella* (4.16% pSS vs. 1.12% HC, *p* = 0.0043, *q*-value = 0.022) and *Streptococcus* (2.54% pSS vs. 1.20% HC, *p* = 0.0043, *q*-value = 0.022), accompanied by a significant decrease in the genera *Bacteroides* (19.47% pSS vs. 26.20% HC, *p* = 0.0017, *q*-value = 0.014), *Alistipes* (3.89% pSS vs. 5.33% HC, *p* < 0.001, *q*-value = 0.006), *Dorea* (0.69% pSS vs. 1.45% HC, *p* = 0.008, *q*-value = 0.038), *Parabacteroides* (2.33% pSS vs. 4.25% HC, *p* = 0.012, Q = 0.046), *Blautia* (2.43% pSS vs. 3.08% HC, *p* = 0.0033, *q*-value = 0.02), *Lachnospira* (1.05% pSS vs. 2.19% HC, *p* = 0.013, *q*-value = 0.05), *Roseburia* (3.29% pSS vs. 6.10% HC, *p* = 0.013, *q*-value = 0.05), *Faecalibacterium* (4.77% pSS vs. 10.97% HC, *p* = 0.0023, *q*-value = 0.02), *Ruminococcus* (2.61% pSS vs. 8.34% HC, *p* = 0.0013, *p* = 0.0013, *q*-value = 0.012) and *Bifidobacterium* (2.72% pSS vs. 4.02% HC, *p* = 0.012, *q*-value = 0.046) (Figure 5).

At the species level, in the pSS group we observed, a significant increase in *Clostridium clostridioforme* (3.05% pSS vs. 2.33% HC, *p* = 0.0037, *q*-value = 0.02), *Escherichia coli* (4.12% pSS vs. 1.45% HC, *p* = 0.0037, *q*-value = 0.02) and *Prevotella copri* (6.78% pSS vs. 2.18% HC, *p* < 0.001, *q*-value = 0.001) and a significant decrease in *Bacteroides fragilis* (7.47% pSS vs. 13.88% HC, *p* = 0.003, *q*-value = 0.0096), *Parabacteroides distasonis* (2.16% pSS vs. 7.37% HC, *p* = 0.0032, *q*-value = 0.027), *Dorea longicatena* (6.97% pSS vs. 4.77% HC, *p* = 0.0045, *q*-value = 0.035), *Ruminococcus lactaris* (2.34% pSS vs. 5.23% HC, *p* = 0.005, *q*-value = 0.047) and *Faecalibacterium prausnitzii* (1.47% pSS vs. 6.72% HC, *p* = 0.0019, *q*-value = 0.02).

### 2.4. Functional Differences in Gut Microbiota between pSS Patients and Healthy Individuals

Kyoto Encyclopedia of Genes and Genomes (KEGG) pathway enrichment analysis of the metagenomic data showed that genes for energy metabolism such as sulfur metabolism (*q*-value = 0.022), carbohydrate metabolism such as butanoate metabolism (*q*-value = 0.019), metabolism of other amino acids such as glutathione metabolism (*q*-value = 0.026), signal transduction mechanisms such as the phosphatidylinositol signaling system (*q*-value = 0.024) and xenobiotics biodegradation and metabolism pathways including benzoate degradation (*q-*value = 0.038) and xylene degradation (*q-*value = 0.04) were significantly depleted in pSS patients compared to healthy control subjects. Nevertheless, in pSS patients compared to healthy controls, there was a significant over-representation of genes in the amino acid metabolism pathways, such as for cysteine and methionine metabolism (*q*-value = 0.029), as well as in genes for membrane transport such as bacterial secretion systems (*q*-value = 0.045), metabolism of cofactors and vitamins such as lipoic acid metabolism (*q*-value = 0.035) and retinol metabolism (*q*-value = 0.027) and glycan biosynthesis and metabolism such as lipopolysaccharide biosynthesis (*q*-value = 0.018) and glycosaminoglycan degradation (*q*-value = 0.023) (Figure 6). 

### 2.5. Differences in the Serum Levels of Inflammatory Mediators and Zonulin and the Relative Expression of FOXP3 in pSS Patients Compared to Healthy Controls

Circulating zonulin levels were significantly higher in pSS patients than in healthy controls (82.26 ± 12.65 ng/mL pSS vs. 35.89 ± 8.94 ng/mL HC, *p* < 0.001). In addition, pSS patients had higher levels of the proinflammatory cytokines IL-6 (199.78 ± 62.91 pg/mL pSS vs. 82.63 ± 18.86 pg/mL HC, *p* < 0.001), IL-17 (5.77 ± 1.79 pg/mL pSS vs. 1.96 ± 0.61 pg/mL HC, *p* < 0.001), IL-12 (65.38 ± 15.95 pg/mL pSS vs. 27.64 ± 10.47 pg/mL HC, *p* < 0.001), TNF-alpha (341.72 ± 52.75 pg/mL pSS vs. 147.48 ± 22.15 pg/mL HC, *p* < 0.001) than healthy controls. Nevertheless, no significant differences were observed in serum levels of IFN-gamma between study groups (2.36 ± 0.95 pg/mL pSS vs. 1.91 ± 0.44 pg/mL HC, *p* = 0.069). With respect to the anti-inflammatory cytokine IL-10 (85.86 ± 26.36 pg/mL pSS vs. 139.89 ± 41.72 pg/mL HC, *p* < 0.001), a significant depletion was found in pSS patients with respect to healthy controls. Finally, relative mRNA expression of FOXP3 in PBMC was significantly lower in pSS patients when compared with healthy controls (0.31 ± 0.11 pSS vs. 1.31 ± 0.23 HC, *p* < 0.001). 

### 2.6. Gut Microbiota was Significantly Associated with Serum Levels of Inflammatory Mediators and Zonulin and the Relative Expression of FOXP3 in pSS Patients

We performed pairwise correlation between the microbial taxonomic composition of both study groups at the different taxa levels with each cytokine and with serum zonulin levels and FOXP3 mRNA expression (Table 2 and Table 3).

Linear regression analysis showed that serum zonulin levels had a significant positive association with the abundance of *Prevotella copri* (R^2^ = 0.752, B = 0.520, *p* = 0.017) and a significant negative association with the abundance of *Bifidobacterium* (R^2^ = 0.752, B = −1.041, *p* < 0.001) in the pSS patients. In the healthy controls, the zonulin level was negatively associated with *Ruminococcus* (R^2^ = 0.752, B = −0.411, *p* = 0.005) and *Bifidobacterium* (R^2^ = 0.752, B = −1.182, *p* < 0.001). 

Similarly, in the pSS patients, the regression analysis showed a negative association between the levels of proinflammatory cytokines IL-17 and TNF-alpha and the abundance of *Bifidobacterium* (R^2^ = 0.742, B = −1.289, *p* < 0.001 and R^2^ = 0.697, B = −0.580, *p* < 0.001, respectively) and *Ruminococcus* (R^2^ = 0.742, B = −0.815, *p* < 0.001) only for IL-17, while the levels of IL-12 were negatively associated with the abundance of *Lachnospira* (R^2^ = 0.897, B = −0.601, *p* < 0.001), *Roseburia* (R^2^ = 0.897, B = −0.641, *p* < 0.001) and *Bifidobacterium* (R^2^ = 0.897, B = −0.715, *p* < 0.001) and positively with the abundance of *Enterobacter* (R^2^ = 0.897, B = 0.562, *p* < 0.001). Concerning the levels of IL-6 in pSS, we found a negative association with *Blautia* (R^2^ = 0.835, B = −0.132, *p* = 0.022) and *Roseburia* (R^2^ = 0.835, B = −0.785, *p* = 0.001) and a positive association with *Escherichia coli* (R^2^ = 0.835, B = 0.678, *p* < 0.001). In the healthy controls, only a significant negative association was found between the levels of the proinflammatory cytokines IL-6 and TNF-alpha and the abundance of *Parabacteroides distasonis* (R^2^ = 0.948, B = 1.139, *p* < 0.001 and R^2^ = 0.570, B = 0.570, *p* = 0.011, respectively)

Conversely, in both the pSS and healthy controls, the anti-inflammatory cytokine IL-10 revealed a strong positive association with the abundance of *Faecalibacterium prausnitzii* (R^2^ = 0.325, B = 0.570, *p* = 0.011; R^2^ = 0.494, B = 1.337, *p* < 0.001, respectively) and *Ruminococcus* (R^2^ = 0.325, B = 0.259, *p* = 0.029; R^2^ = 0.494, B = 0.259, *p* = 0.029, respectively). Finally, FOXP3 mRNA expression was positively associated with the abundance of *Bacteroides fragilis* in both the pSS patients (R^2^ = 0.547, B = 0.548, *p* = 0.029) and healthy controls (R^2^ = 0.547, B = 0.670, *p* = 0.016).

## 3. Discussion

This study has demonstrated that the pSS patients had gut dysbiosis associated with increased serum levels of proinflammatory cytokines including IL-6, IL-12, IL-17 and TNF-alpha (systemic inflammation) and zonulin (intestinal permeability) that resulted in increased systemic microbial exposure. The pSS patients included were from the same geographical area and had a similar BMI, sex, age and ethnic background as the healthy controls to avoid the influence of these confounding factors on the gut microbiota analysis.

Microbiota diversity is essential to maintaining ecosystem stability and efficiency. Regarding alpha diversity, patients exhibited decreased gut microbiota diversity and richness compared with controls. Van der Meulen et al. also found significantly lower richness in gut microbiota composition but not in diversity in pSS patients compared to healthy controls [18]. Reduced diversity may favor emergence of pathogenic bacteria that disrupt the intestinal barrier and stimulate production of inflammatory mediators by mucosal epithelial cells in the intestinal lamina propria and mesenteric lymph nodes [7]. Moreover, individuals with low gut microbiota richness have higher inflammatory markers in blood [19]. These data could suggest a link between the lower bacterial diversity and richness and the higher systemic inflammation and intestinal permeability found in our pSS patients. In addition, the Bray-Curtis PCoA plot for beta diversity showed that the pSS patients were grouped into a tight cluster compared to the healthy controls, indicating significant disease-mediated microbial changes. 

We found large significant differences in predominant phyla, family, genera and species taxa in pSS patients compared to healthy controls. At the genera level, gut dysbiosis showed an increase in the abundance of the mucin-degrading and enteric pathogens *Prevotella, Clostridium, Enterobacter, Escherichia*, and *Streptococcus* and a depletion in the relative abundance of *Bacteroides, Parabacteroides, Faecalibacterium, Roseburia, Ruminococcus, Dorea, Alistipes, Blautia* and *Bifidobacterium*. Our findings have demonstrated both similarities and differences compared to prior data in Sjögren’s patients. de Paiva et al. showed that fecal samples from Sjögren’s patients had a reduction in the genus *Bacteroides, Parabacteroides* and *Faecalibacterium* and an augmentation of *Escherichia* and *Streptococcus* genera compared to controls. Nevertheless, unlike our study these authors described a significant decrease in the abundance of *Prevotella* in the gut of Sjögren’s patients [7]. 

Conversely, but similar to our data, in a very recent study, Mendez et al. reported a decrease in the relative abundance of *Faecalibacterium* and *Bacteroides* and an increase in the relative abundance of *Prevotella* in Sjögren’s patients [17]. *Prevotella* is a crucial genus in the onset of other chronic inflammatory disorders such as rheumatoid arthritis [20]. Sher et al. identified a strong association between *Prevotella copri* and new-onset untreated rheumatoid arthritis and the higher abundance of *Prevotella* was associated with a reduction in several beneficial microbes, including *Bacteroides* [21].

*Bacteroides* spp. and the Clostridia clusters XIVa and IV (butyrate-producing bacteria) have been shown to be necessary for maintaining the balance between anti-inflammatory regulatory Treg cells and proinflammatory Th17 cells, which protect the mucosa from pathogenic microorganism colonization [22]. The reduction found in our study in both butyrate-producing bacteria and *Bacteroides* may affect the balance between Th17 and Treg cells in pSS patients, contributing to autoimmunity.

In another recent study, Moon et al. demonstrated that SS patients compared to healthy controls had increased Bacteroidetes levels and a decreased Firmicutes/Bacteroidetes ratio, while Actinobacteria (*Bifidobacterium*) and Clostridia (*Blautia, Dorea* and *Agathobacter*) were significantly decreased [23]. The depletion in the abundance of *Bifidobacterium* in pSS patients shown in our study and by other authors is also described in other chronic inflammatory states including rheumatoid arthritis and Crohn’s disease [9,24]. Moreover, Mandl et al. indicated that SS patients with severe dysbiosis and decreased levels of *Bifidobacterium* and *Alistipes* could have higher disease activity, lower levels of complement component and higher levels of fecal calprotectin [8]. The possible causes of these differences in gut microbiota composition between studies could be differences in dietary habits or in the severity of Sjögren’s syndrome, medications, age or sex bias. Several studies have described age- and sex-related changes in human gut microbiota composition [25,26,27].

On the other hand, this gut microbiota dysbiosis also affects the imbalance of T helper 1 and Th17 cells, the polarization of Tregs, the increased permeability of intestinal epithelial cells and the production of short-chain fatty acids [9]. In the pSS patients, we found a significant positive association between the abundance of *Prevotella copri* and serum levels of zonulin. Zonulin is a protein, synthesized in intestinal and liver cells, that reversibly modulates the permeability of the intestinal epithelial barrier by disassembling intercellular tight junctions [28,29]. Wright et al. found that *Prevotella* contains enzymes that are important in mucin degradation, which may disrupt the colonic mucus barrier and increase intestinal permeability [30], allowing the diffusion of pathogens, toxins, and antigens from the luminal environment into the mucosal tissues and circulatory system [31], resulting in the immune activation and tissue inflammation important in the onset or progression of several intestinal and chronic autoimmune diseases [32]. Conversely, a recent study in immunocompetent mice has suggested that alteration of the gut microbiome by *Prevotella* spp. colonization decreases levels of short chain fatty acids, especially acetate, enhancing intestinal inflammation but after chemical damage to the intestinal barrier [33]. Accordingly, the significant increase in *Prevotella* abundance found in our study could be associated with the immune dysregulation and the increased intestinal inflammation in the pSS patients.

In this study, pSS patients had significantly higher levels of the proinflammatory cytokines IL-17, IL-12, IL-6 and TNF-alpha along with a significant decrease in the anti-inflammatory cytokine IL-10 and FOXP3 mRNA expression compared with healthy controls. Moreover, the significant correlations found between gut microbiota and host serum levels of proinflammatory and anti-inflammatory cytokines and FOXP3 mRNA expression indicate that the significant depletion of *Roseburia*, *Blautia*, *Lachnospira, Ruminococcus, Bifidobacterium, Faecalibacterium prausnitzii* and *Bacteroides fragilis* able to exert anti-inflammatory effects (through cytokine production) and improve gut barrier function (through butyrate production) [34,35], together with the significant increase in *Enterobacter* and *Escherichia coli* (potential enteric pathogens with proinflammatory capacity) could increase intestinal permeability and low-grade inflammation in pSS patients. Butyrate (a gut microbiota-derived metabolite) is a crucial metabolite providing energy for colonic epithelial cells to maintain intestinal barrier functions. Moreover, butyrate acts as an anti-inflammatory molecule, capable of inhibiting NF-κB activation in the host immune cells by binding to G protein-coupled receptors (GPR43 and GPR41), thereby blocking inflammatory responses and suppressing TNF-alpha and IL-6 release [36]. Butyrate also leads to modulation of IL-17 expression [37] and promotes Treg cell differentiation, which can ultimately suppress proinflammatory responses [38]. Fernando et al. revealed that butyrate down-regulates expression of pathologic levels of IL-17 [39]. 

Similarly, Säemann et al. demonstrated that butyrate is able to strongly inhibit the production of proinflammatory cytokines IL-12 and TNF-alpha by monocytes, whereas the anti-inflammatory cytokine IL-10 is significantly increased after bacterial stimulation [40]. Finally, the, activation of Tregs by butyrate not only inhibits effector T cells but also produces the anti-inflammatory cytokine IL-10 [41]. The immunomodulatory molecule polysaccharide A derived from the human commensal *Bacteroides fragilis* mediates the conversion of CD4+ T cells into FOXP3+ Treg cells that produce IL-10 during commensal colonization, stimulating immunological development within mammalian hosts [42]. These data may indicate the possible association between butyrate-related immune dysregulation and alterations in gut barrier function in pSS.

Our study also explored the functionality of the gut microbiota. Phylogenetic Investigation of Communities by Reconstruction of Unobserved States (PICRUSt) analysis showed that bacterial metabolic pathways were also different between pSS patients and healthy controls. The gut microbiome of pSS patients showed a depletion in genes involved in metabolic functions such as butanoate metabolism and glutathione metabolism as well as an over-representation of genes necessary for lipoic acid metabolism, retinol metabolism, lipopolysaccharide biosynthesis and glycosaminoglycan degradation. The depletion in genes involved in butanoate metabolism in the pSS patients could be due to the significantly lower abundance of the families Ruminococcaceae (in particular *Faecalibacterium prausnitzii)* and Lachnospiraceae *(especially Roseburia* spp.), the main butyrate producing-bacteria belonging to the phylum Firmicutes, despite the significantly higher abundance in these patients of other potential butyrate producers such as Clostridiaceae, Veillonellacea and Prevotellaceae. Clostridiaceae, Prevotellaceae and Veillonellaceae comprise mainly bacteria that produce acetate, succinate and propionate, although some microorganisms of these families could promote butyrate production via acetate or succinate interconversion [43,44]. The decrease in genes related to butyrate synthesis due to the decrease in beneficial butyrate-producing bacteria may predispose to increased intestinal permeability associated with autoimmune conditions [45]. In systemic lupus erythematosus patients, depletion of glutathione has been associated with various immune abnormalities including deregulation of apoptosis and abnormal cytokine and chemokine production [46]. Microbial pathogens implement basic lipoylation strategies, which can affect their pathogenesis and virulence [47]. Moreover, lipoic acid metabolism has been thoroughly characterized in the Gram-negative bacterium *Escherichia coli*, which is significantly increased in pSS patients with respect to controls in our study. Indeed, glycosaminoglycans help to form a protective barrier of intestinal mucin, and the breakdown of glycosaminoglycans have been reported to be associated with an inflammatory response in inflammatory bowel disease [48]. Retinoic acid (a product of retinol metabolism) mediates cellular regulation in autoimmune diseases through suppression of inflammatory T helper 1/Th17 responses by decreasing IFN-gamma and IL-17 [49]. All these data suggest that the gut microbiota of the pSS patients was enriched in genes from pathways associated with bacterial pathogenesis and potentially related to chronic inflammation compared to healthy controls.

Findings from this study should be interpreted bearing in mind our study limitations which include the small population and the gender bias of the disease. In addition, our results should be confirmed by studies in larger populations with both female and male patients that mirror the general pSS patient population.

## 4. Materials and Methods

### 4.1. Study Subjects

A total of 19 pSS patients between the ages of 18 and 75 years were recruited at the Ophthalmology Service at Arruzafa Hospital. All recruited pSS patients met the American College of Rheumatology/European League Against Rheumatism (ACR-EULAR) classification criteria [50] and were subjected to a complete ocular, oral, and rheumatologic evaluation. The exclusion criteria were: pregnancy, use of local or systemic antibiotics or probiotics 3 months prior to enrollment in the study, the presence of secondary Sjögren’s syndrome or other autoimmune disease, and concurrent inflammatory bowel disease.

Additionally, 19 age-, sex- and BMI-matched healthy controls were enrolled in the study. The exclusion criteria for the healthy controls were: diagnosis of gut disease and use of antibiotics or probiotics within the past 3 months before sampling.

All pSS patients and healthy controls included in the study followed the Mediterranean diet (rich in olive oil, nuts, fruits, vegetables, legumes, whole grains and fish, with a low intake of meat and dairy products). Adherence to the Mediterranean diet was assessed via a validated food frequency questionnaire, composed of 14 questions regarding the main food groups consumed as part of the Mediterranean diet [51].

The study protocol (2182-N-20) was approved by the Medical Ethics Committee of Arruzafa Hospital (1 July 2019) and conducted in accordance with the Declaration of Helsinki for experiments involving human subjects. All participants enrolled were verbally informed of the characteristics of the study and written informed consent was obtained. 

### 4.2. Laboratory Measurements

Peripheral venous blood samples were obtained from the study subjects after 12 h of fasting. Serum was centrifuged and frozen at −80 °C until analysis. Levels of fasting glucose, total cholesterol, triglycerides, HDL-cholesterol and LDL-cholesterol were analyzed by enzymatic methods (Randox Laboratories Ltd., Crumlin, UK) using a Dimension autoanalyzer (Dade Behring Inc., Deerfield, IL, USA). The autoantibodies anti-Ro/SSA and anti-La/SSB were quantified using a multiplex immunobead assay (Quanta Plex SLE Profile 8; Inova Diagnostics) in a Luminex 100 instrument applying the cut-off values recommended by the manufacturer. Antinuclear autoantibodies (ANA) were measured by indirect immunofluorescence technique employing HEp2 cells (Biorad, Marnes la Coquette, France). The rheumatoid factor and complement component 3 (C3) and complement component 4 (C4) were analyzed by a nephelometry technique (BNII nephelometer; Dade Behring, Marburg, Germany).

### 4.3. DNA Extraction and Gut Microbiota Sequencing

DNA was extracted from 200 mg of stool samples using the QIAamp DNA stool Mini kit (Qiagen, Hilden, Germany) according to the manufacturer’s protocol. DNA concentration was measured and the purity was verified with a Nanodrop spectrophotometer (Nanodrop Technologies, Wilmington, DE, USA). The Ion 16S Metagenomics Kit (Thermo-Fisher Scientific Inc., Waltham, MA, USA) was used to amplify the 16S rRNA gene region from stool DNA using two primer pools (V2–4–8 and V3–6, 7–9) covering hypervariable regions of the 16S rRNA region in bacteria. The Ion Plus Fragment Library Kit (Thermofisher Scientific Inc, Waltham, MA, USA) was used to ligate barcoded adapters to the generated amplicons and create the barcoded libraries. Template preparation of the created amplicon libraries was done on the automated Ion Chef System using the Ion 520TM/530TM Kit-Chef (Thermo Fisher Scientific, Waltham, MA, USA) according to the manufacturer’s instructions. Sequencing was carried out on an Ion 520 chip employing the Ion S5 platform (Thermofisher Scientific Inc., Waltham, MA, USA).

### 4.4. Bioinformatics Analysis

Quantitative Insights into Microbial Ecology (QIIME2, version 2019.4) software [52] was used to analyze sequence quality and for diversity and taxonomic analysis as previously described [53]. Alpha (Shannon and Chao1) and beta (Bray-Curtis dissimilarity) diversity metrics were estimated with the Kruskal—Wallis test and the permutational multivariate analysis of variance (PERMANOVA) with 999 permutations respectively. Taxonomic analysis was done by clustering with VSEARCH and the reference base Greengenes version 13_8 at 97% of identity.

Metagenome function was predicted by Phylogenetic Investigation of Communities by Reconstruction of Unobserved States (PICRUSt) analysis by picking OTUs against the Greengenes database [54]. Statistical analyses were performed in R 3.6.0 using the R package “pheatmap” for data analysis and plotting. P-values were corrected for multiple comparisons using the Benjamini–Hochberg method. A corrected *p* < 0.05 was considered to be statistically significant.

### 4.5. Intestinal Permeability Analysis

The analysis of serum zonulin was performed using a Zonulin ELISA kit (Immundiagnostik AG, Bernsheim, Germany; intra and inter-assay variation was between 3–10% with a detection limit of 0.22 ng/mL) according to the manufacturer’s protocol. Standards and study samples were tested in duplicate.

### 4.6. Measurement of Serum Cytokine Levels

Concentrations of IL-6, IL-12, IL-17, TNF-alpha and IFN-gamma were quantified by ELISA assay kits (Thermo Fisher Scientific, MA, USA) in serum samples according to the manufacturer’s instructions. Detection limits were: 7.8–500 pg/mL for IL-6, 1.56–100 pg/mL for IL-12, 0.23–15 pg/mL for IL-17, 15.6–1000 pg/mL for TNF-alpha and 1.6–100 pg/mL for IFN-gamma. IL-10 cytokine was measured by Novex^®^ ELISA Kits (Life Technology, ES, Carlsbad, CA, USA), performed according to the manufacturer’s instructions. The assay range was 7.8–500 pg/mL for IL-10. 

### 4.7. Real-Time Quantitative PCR for Treg Cells

The total RNA from peripheral blood mononuclear cells was extracted using a QIAamp RNA Blood Mini Kit following the manufacturer’s protocol. The concentration of RNA was measured with a spectrophotometer (Nanodrop N-100, Thermo Scientific, Waltham, MA, USA) and cDNA was synthesized using the Transcriptor First Strand cDNA Synthesis Kit (Roche) and random hexamers. Real time PCR was performed using a specific pre-validated and commercially available TaqMan^®^ primer/probe set for the FOXP3 gene (RefSeq NM_001114377.1) in an ABI 7500 Fast Real-Time PCR System (Applied Biosystems, Foster City, CA, USA). The 18S rRNA (4319413E) gene was used as the endogenous reference in each reaction. The relative *FOXP3* gene expression levels were calculated using the 2^−∆∆CT^ method [55]. All tests were done in duplicate.

### 4.8. Statistical Analysis

The Kruskal–Wallis rank-sum test was used to compare differential abundances of taxa between the pSS patients and healthy individuals, and the false discovery rate (FDR) using the Benjamini–Hochberg method was applied to correct the significant *p*-values (*q*-value < 0.05). 

Anthropometric and clinical characteristics were analyzed with SPSS Statistics V.20.0 (SPSS, Chicago, IL, USA). Continuous variables were compared between groups using the Mann–Whitney U test, while categorical variables were compared by Fisher’s exact test. The relationship between gut microbiota and study variables was analyzed using Spearman correlation models. A linear regression analysis was performed to identify individual bacteria as independent predictors for levels of inflammatory mediators, serum zonulin and expression of FOXP3 in the study groups. The results were presented as mean ± standard deviation (SD) for quantitative variables and as frequencies and percentages for qualitative variables. Statistical significance was established at *p* < 0.05. 

## 5. Conclusions

Our data suggest that the gut microbiota in individuals with pSS differs at the taxonomic and functional levels in comparison with healthy controls. pSS was associated with significantly lower microbiota diversity and richness, a significantly higher relative abundance of opportunistic pathogens with proinflammatory activity (such as *Prevotella, Streptococcus*, *Enterobacter* and *Escherichia coli*) and a lower relative abundance of beneficial or commensal butyrate-producing bacteria (such as *Faecalibacterium prausnitzii*, *Roseburia*, *Blautia*, *Lachnospira*, *Ruminococcus*, *Bacteroides fragilis* and *Bifidobacterium*). Moreover, pSS patients had significantly higher levels of proinflammatory cytokines (IL-6, IL-12, IL-17 and TNF-alpha) and serum zonulin, and significantly lower levels of IL-10 and FOXP3 mRNA expression (implicated in the development and function of Treg cells) than healthy controls. The PICRUSt analysis found a depletion in genes involved in metabolic functions such as butanoate metabolism and glutathione metabolism as well as an over-representation of genes involved in lipoic acid metabolism, retinol metabolism, lipopolysaccharide biosynthesis and glycosaminoglycan degradation, genes from pathways associated with bacterial pathogenesis and potentially related to chronic inflammation. This data provides novel and precise diagnostic biomarkers based on gut microbiota as well as new foundations for the design of new microbe-based therapies for Sjögren’s syndrome.

## Figures and Tables

**Figure 1 ijms-21-08733-f001:**
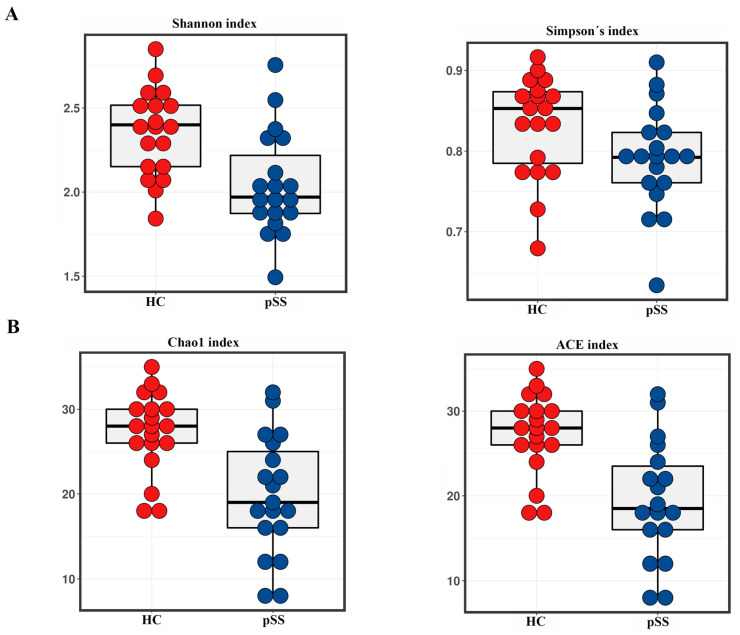
Comparison of alpha diversity in primary Sjögren’s syndrome (pSS) patients and healthy controls (HC). (**A**) Shannon and Simpson’s diversity indices; (**B**) Chao1 and ACE richness indices.

**Figure 2 ijms-21-08733-f002:**
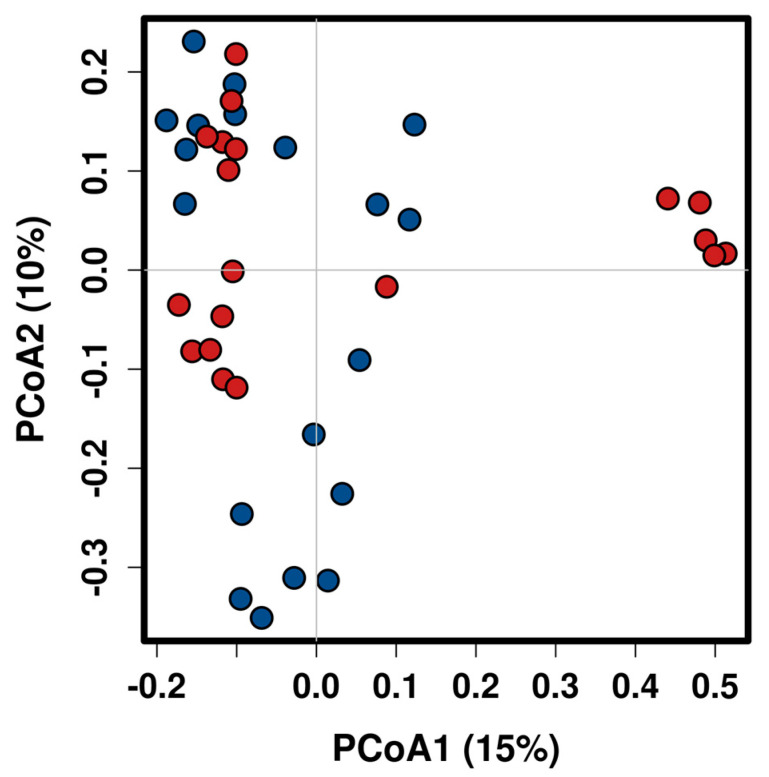
Principal Coordinates Analysis (PCoA) of bacterial communities from primary Sjögren’s syndrome (pSS) patients and healthy controls using Bray-Curtis similarity index at the genus level. Blue dot (pSS patients) and red dot (healthy controls).

**Figure 3 ijms-21-08733-f003:**
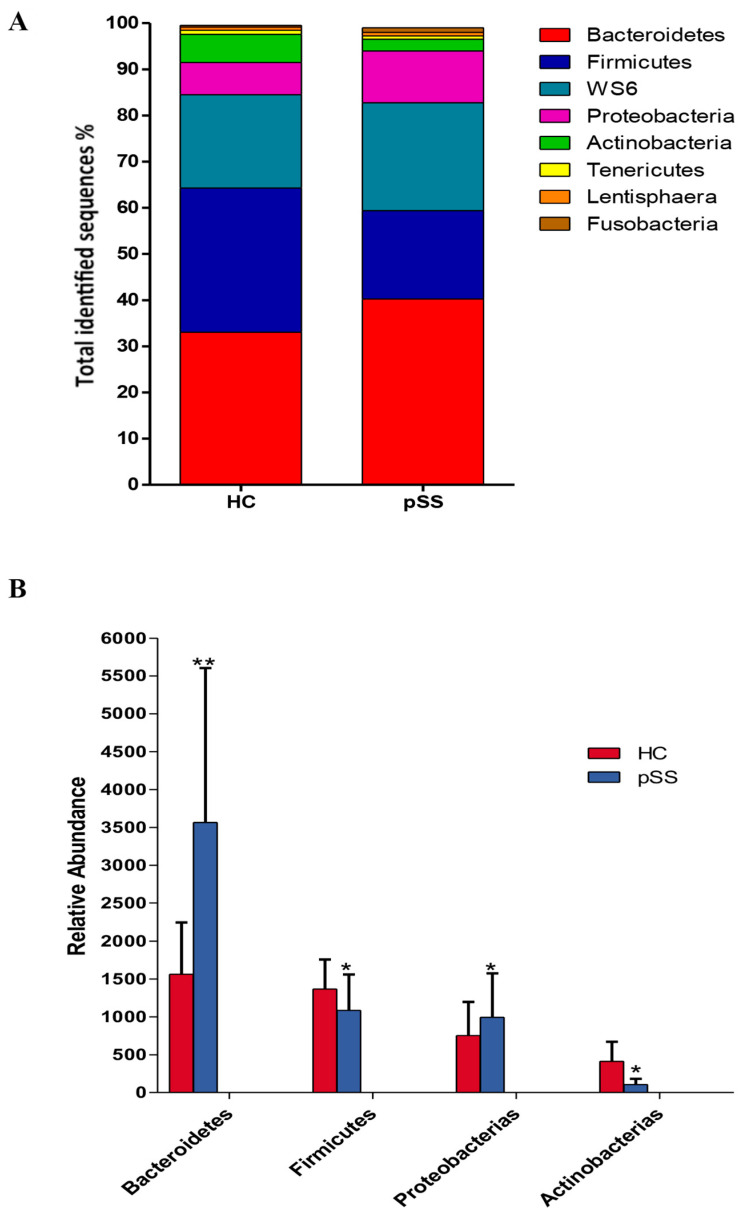
Composition of fecal microbiota at the phylum levels in primary Sjögren’s syndrome (pSS) patients and healthy controls (HC). (**A**) Data are shown as a percentage of the total identified sequences per group. (**B**) Differentially abundant phyla in the stool samples of pSS patients compared to HC * *p* < 0.05, ** *p* <0.001. The bars indicate mean ± standard deviation (SD).

**Figure 4 ijms-21-08733-f004:**
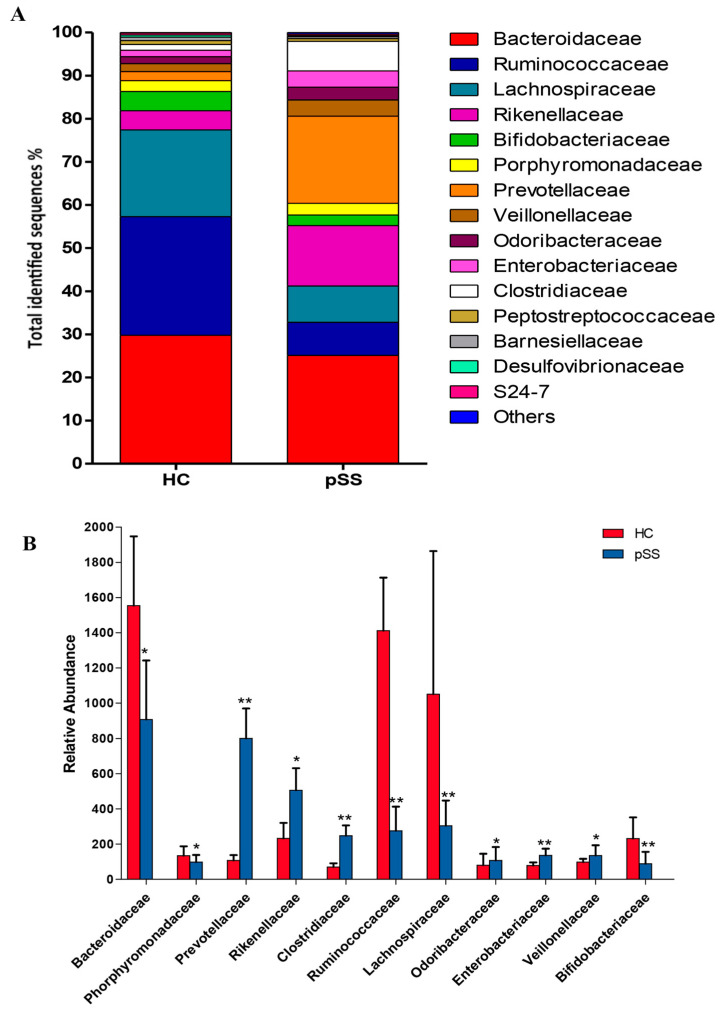
Family-level microbial classification of bacteria from stool samples of primary Sjögren’s syndrome (pSS) patients and healthy controls (HC). (**A**) Data are shown as a percentage of the total identified sequences per group. (**B**) Differentially abundant families in the stool samples of pSS patients compared to HC. * *p* < 0.05, ** *p* < 0.001. The bars indicate mean ± standard deviation (SD).

**Figure 5 ijms-21-08733-f005:**
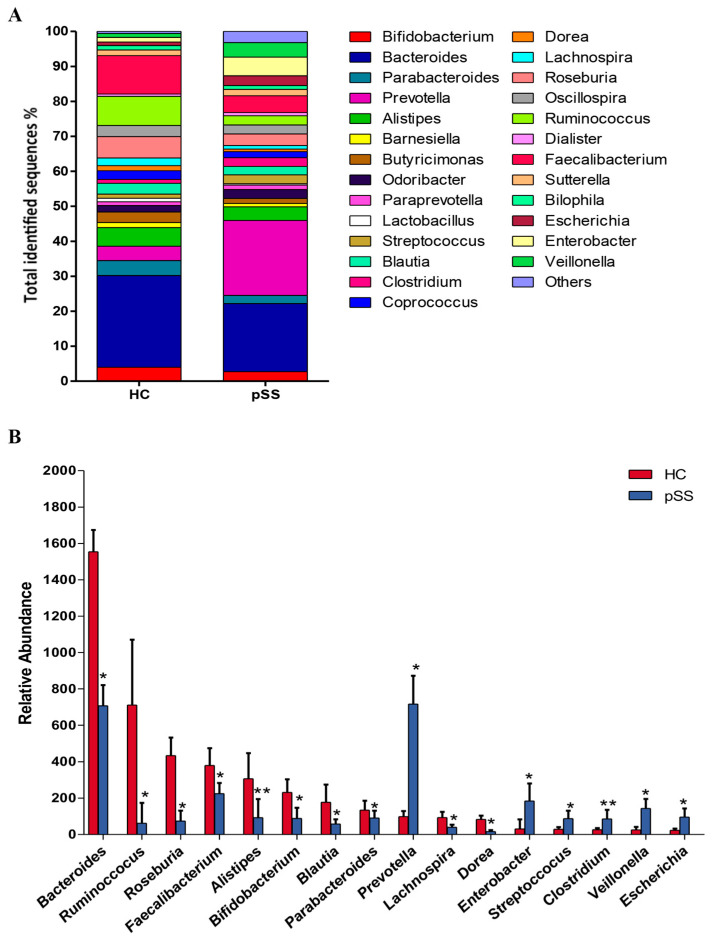
Relative abundance of bacterial genera in the microbiota of primary Sjögren’s syndrome (pSS) patients and healthy controls (HC). (**A**) Data are shown as a percentage of the total identified sequences per group. (**B**) Differentially abundant genera in the stool samples of pSS and HC. * *p* < 0.05, ** *p* < 0.001. The bars indicate mean ± standard deviation (SD).

**Figure 6 ijms-21-08733-f006:**
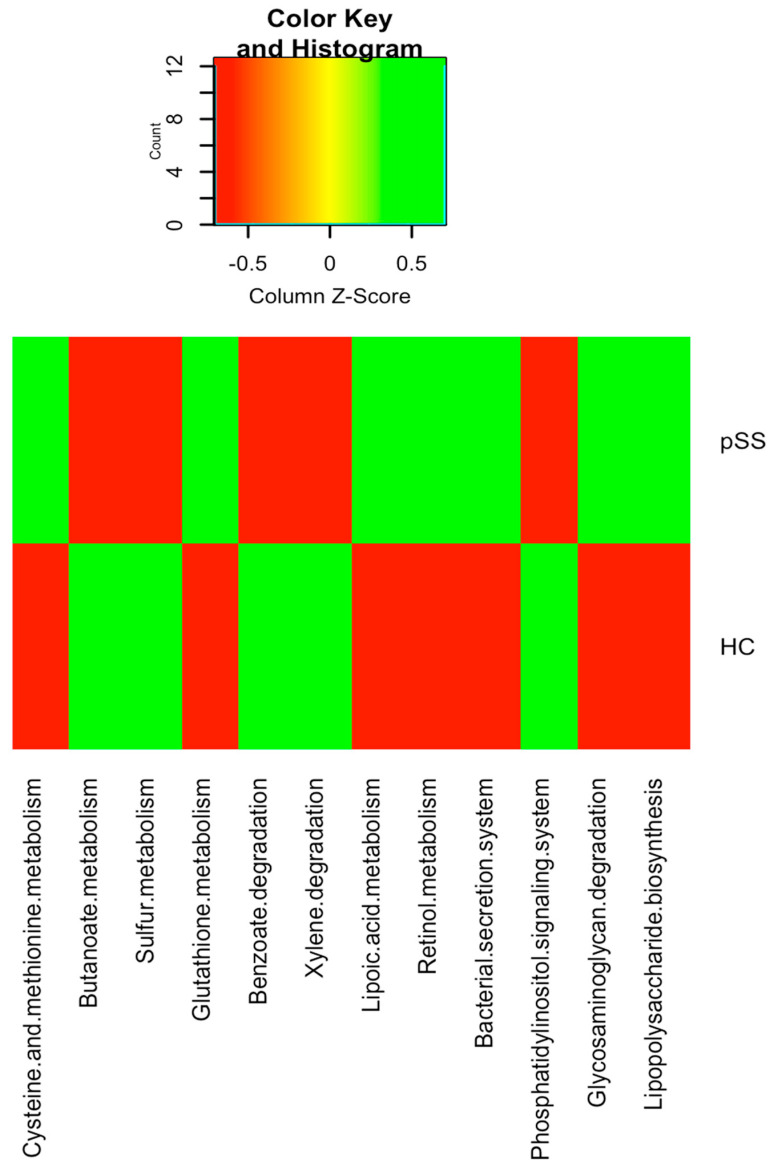
Heatmap of bacterial gene functional predictions using the Phylogenetic Investigation of Communities by Reconstruction of Unobserved States (PICRUSt) algorithm from fecal samples from primary Sjögren’s syndrome (pSS) patients and healthy controls (HC).

**Table 1 ijms-21-08733-t001:** Demographic and clinical characteristics of primary Sjögren’s syndrome (pSS) patients and healthy controls.

	pSS (*n* = 19)	Healthy Controls (*n* = 19)	*p*
Age in years, mean ± SD	56.43 ± 8.74	54.70 ± 8.42	0.545
Female sex, n (%)	19 (100)	19 (100)	0.589
BMI, mean ±SD	26.17 ± 6.27	26.25 ± 4.15	0.963
Smoking, n (%)	7 (36.84)	8 (42.10)	0.736
Caucasian, n (%)	19 (100)	19 (100)	0.589
Fulfilling ACR EULAR 2016 criteria for SS, n (%)	19 (100)	NA	-
**Serum biochemistry**			
Glucose (mg/dl)	93.44 ± 22.14	92.16 ± 18.24	0.847
Insulin (mg/dl)	8.81 ± 3.63	6.82 ± 3.72	0.104
Triglycerides (mg/dl)	90.41 ± 29.65	97.92 ± 27.51	0.398
Cholesterol (mg/dl)	189.38 ± 22.29	176.3 ± 34.6	0.175
LDL-cholesterol (mg/dl)	123.76 ± 31.0	118.20 ± 33.75	0.600
HDL-cholesterol (mg/dl)	63.16 ± 14.73	59.84 ± 15.60	0.504
CRP (ng/mL)	3.80 ± 0.49	2.77 ± 0.63	0.001
ANA positive, n (%)	15 (78.94)	NA	-
ENA positive, n (%)	6 (31.57%)	NA	-
Ro/SSA positive, n (%)	11 (57.89)	NA	-
La/SSB positive, n (%)	12 (63.15)	NA	-
Rheumatoid factor positive, n (%)	13 (68.42)	NA	-
C3 low at inclusion (<0.90 g/L), n (%)	7 (36.84)	NA	-
C4 low at inclusion (<0.10 g/L), n (%)	7 (36.84)	NA	-
**Other symptoms reported**			
Hypothyroidism, n (%)	3(15.78)	0 (0)	-
Fibromyalgia, n (%)	2 (10.52)	0 (0)	-
Oral ulcers, n (%)	1 (5.26)	0 (0)	-
Arthralgia, n (%)	3 (15.78)	0 (0)	-
Type 2 diabetes, n (%)	1 (5.26)	0 (0)	-

pSS, primary Sjögren’s syndrome; BMI, body mass index; CRP, c-reactive protein; LDL, Low density lipoprotein-cholesterol; HDL-cholesterol, high density lipoprotein-cholesterol; ACR EULAR, American College of Rheumatology/European League Against Rheumatism; ANA, antinuclear antibody; ENA, extractable nuclear antigen; Ro/SSA, anti-Ro/Sjögren’s syndrome autoantibody A; La/SSB, anti La/Sjögren’s syndrome autoantibody B; C3/C4, complement C3/C4.

**Table 2 ijms-21-08733-t002:** Correlations between gut microbiota composition and serum levels of IL-6, IL-12 and IL-17 in the primary Sjögren’s syndrome (pSS) and healthy control groups.

	pSS	Healthy Controls	pSS	Healthy Controls	pSS	Healthy Controls
	IL-6		IL-12		IL-17	
*Butyciromonas*	−0.598 (*p* = 0.030)					
*Roseburia*	−0.718 (*p* = 0.010)		−0.767 (*p* = 0.020)		−0.511 (*p* = 0.045)	−0.675 (*p* = 0.047)
*Ruminococcus*	−0.527 (*p* = 0.020)	−0.533 (*p* = 0.029)			−0.631 (*p* = 0.025)	
*Blautia*	−0.635 (*p* = 0.007)					
*Enterobacter*			0.873 (*p* = 0.008)		0.549 (*p* = 0.039)	
*Streptococcus*			0.466 (*p* = 0.044)			
*Faecalibacterium prausnitzii*		−0.587 (*p* = 0.028)	−0.508 (*p* = 0.027)	−0.550 (*p*= 0.025)		
*Escherichia coli*	0.723 (*p* = 0.009)					
*Bifidobacterium*		−0.573 (*p* = 0.030)	−0.508 (*p* = 0.018)	−0.507 (*p* = 0.027)	−0.789 (*p* = 0.002)	−0.459 (*p* = 0.041)
*Bacteroides fragilis*					−0.473 (*p* = 0.041)	
*Parabacteroides distasonis*		−0.767 (*p* = 0.004)				
*Lachnospira*			−0.514 (*p* = 0.014)			
*Prevotella*	0.465 (*p* = 0.045)					
*Clostridium*	0.462 (*p* = 0.046)				0.412 (*p* = 0.043)	

**Table 3 ijms-21-08733-t003:** Correlations between gut microbiota composition and serum levels of TNF-alpha, IL-10, zonulin and FOXP3 mRNA expression in the study groups.

	pSS	Healthy Controls	pSS	Healthy Controls	pSS	Healthy Controls	pSS	Healthy Controls
	TNF-Alpha	IL-10	Zonulin	FOXP3 mRNA Expression
*Ruminococcus*		−0.496 (*p* = 0.031)	0.563 (*p* = 0.035)	0.494 (*p* = 0.032)	−0.496 (*p* = 0.031)	−0.665 (*p* = 0.005)		0.573 (*p* = 0.034)
*Butyciromonas*	−0.560 (*p* = 0.033)		0.525 (*p* = 0.021)			−0.542 (*p* = 0.027)	0.598 (*p* = 0.043)	
*Blautia*			0.516 (*p* = 0.024)	0.576 (*p* = 0.005)				
*Roseburia*					−0.458 (*p* = 0.048)			0.501 (*p* = 0.042)
*Prevotella copri*					0.645 (*p* = 0.003)			
*Clostridium*	0.550 (*p* = 0.025)		−0.651 (*p* = 0.033)					
*Bifidobacterium*	−0.580 (*p* = 0.009)				−0.631 (*p* = 0.010)	−0.584 (*p* = 0.017)		
*Bacteroides fragilis*	−0.567 (*p* = 0.041)						0.631 (*p* = 0.002)	0.563 (*p* = 0.009)
*Faecalibacterium prausnitzii*			0.651 (*p* = 0.010)	0.597 (*p* = 0.025)		−0.502 (*p* = 0.029)		
*Parabacteroides distasonis*		−0.559 (*p* = 0.009)				−0.700 (*p* = 0.035)

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
