# Peer review of "Connection between the Gut Microbiome, Systemic Inflammation, Gut Permeability and FOXP3 Expression in Patients with Primary Sjögren’s Syndrome"

_ijms, 2020, doi:10.3390/ijms21228733_

Round 1

Reviewer 1 Report

Study summary

The manuscript by Cano-Ortiz et al. compares the fecal microbiota signature in 19 patients with primary Sjögren syndrome and 19 healthy controls. The authors aimed to determine whether there was a relationship between microbial composition and inflammation, markers of functional regulatory T cell and of intestinal permeability. Results revealed specific microbial changes, e.g. reduction in Firmicutes and Actinobacteria, and a rise in Bacteroidetes and Proteobacteria, associated with immune dysregulation and proinflammatory status.

Overall, the topic of the study is not particularly original, and its relevance lies mainly in the fact that it provides further evidence of altered gut microbial composition in Sjögren's syndrome. Analyses were conducted in a correct way, the statistical analysis is acceptable, relevant results are reported and properly interpreted, and conclusions are adequately summarized.  English was well written, although it could be improved in some sentences. The quality of the figures and tables is overall satisfactory. The reference list is updated and covers the relevant literature, but contains many errors.

Comments and suggestions to the Authors

Lines 83-84. The authors reported that no patient had autoimmune comorbidities, yet there were 3 hypothyroid patients, so I presume it wasn't Hashimoto's thyroiditis.  Similarly for the case of diabetes that I presume was not autoimmune.

Table 1. All study participants were women, likely due to the small sample size and gender bias of the disease. This may represent a limitation to the generalizability of the study, which must be duly mentioned in the discussion.

Lines 283-285. At functional level the authors state that genes for butanoate metabolism were significantly depleted in pSS patients’ microbiota. How can this result be reconciled with the increased abundance of Clostridiaceae and other butyrate-producers (Figure 4)? Please add a comment in the discussion.

Lines 294-295 and 364-366. Was a vegan/vegetarian diet among the exclusion criteria of the study? This could especially be important in relation to the result relating to the Prevotella genus. An increased Prevotella in the intestine has been associated with the consumption of a plant- based diet (PMID: 26229982 and PMID: 25892234). This information should be added in the method and result sections or, if not available, added as a limitation in the discussion.

Lines  298-299. Experimental studies indicate that gut colonization by Prevotella does not in itself induce disruption to the intestinal barrier, but can lead to immune dysregulation when the barrier is damaged by other factors (see also PMID: 32433514). Hence, Prevotella could only be an enterotype marker without pathogenic role (PMID: 21508958).

Line 446. Statistical Analysis. The authors reported that the comparison between two groups was made with the Wilcoxon rank-sum test, and later, on lines 449-450, mention is made of the Mann-Whitney U test. Since the two names designate the same test, it is preferable to use only one wording. Please note that the Wilcoxon rank-sum test is not the Wilcoxon Signed Rank Test.

Minor points

Line 78. I suggest: “clinical characteristics”.

Table 1. Please indicate insulin concentration units.

Figure 1. In general, the scatter points are too large causing overlap between them. A slight reduction in size may be advisable.

Figure 3, 4 and 5. In the y axis, please correct “identificed” to “identified”.

Figure 4. Please double check the names of the families in the x axis (eg. Bacteroidaceae).

Line 137. I suggest to replace “higher” with “more frequent”.

Line 212. I suggest: “was significantly associated”.

Lines 355-357: These sentences are template instructions and must be deleted.

Line 440: Perhaps the authors meant 2−ΔΔCT method ?

Line 451: I suggest "linear regression analysis".

References are not formatted correctly or include errors in the authors’ name. Please check all references accurately.

Line 511. Reference no. 1. The authors are: Mariette X, Criswell LA.

Line 512, line 514 and line 605. Please add the umlaut “ö” to the surname Sjögren.

Line 517. The first author’s surname is Breban while Maxime is the name.

Line 521. The third author's name must have an acute accent on the “o”.

Line 537. Please add an umlaut “ä” to the surname of the seventh author.

Line 544. The surname of the third author is Ramirez Vinolo not just Vinolo. I also believe the last author's surname is “dos Santos Martin”.

Line 584. The abbreviated title of the journal is: Am J Physiol Gastrointest Liver Physiol.

Author Response

The manuscript by Cano-Ortiz et al. compares the fecal microbiota signature in 19 patients with primary Sjögren syndrome and 19 healthy controls. The authors aimed to determine whether there was a relationship between microbial composition and inflammation, markers of functional regulatory T cell and of intestinal permeability. Results revealed specific microbial changes, e.g. reduction in Firmicutes and Actinobacteria, and a rise in Bacteroidetes and Proteobacteria, associated with immune dysregulation and proinflammatory status.

Overall, the topic of the study is not particularly original, and its relevance lies mainly in the fact that it provides further evidence of altered gut microbial composition in Sjögren's syndrome. Analyses were conducted in a correct way, the statistical analysis is acceptable, relevant results are reported and properly interpreted, and conclusions are adequately summarized.  English was well written, although it could be improved in some sentences. The quality of the figures and tables is overall satisfactory. The reference list is updated and covers the relevant literature, but contains many errors.

Response:First of all, we thank the reviewer for his/her comments and for offering us a constructive review of our manuscript.

Comments and suggestions to the Authors

Comment 1:Lines 83-84. The authors reported that no patient had autoimmune comorbidities, yet there were 3 hypothyroid patients, so I presume it wasn't Hashimoto's thyroiditis.  Similarly for the case of diabetes that I presume was not autoimmune.

Response 1:As indicated by the reviewer, in our study, 3 patients had hypothyroidism but not Hashimoto's thyroiditis, and 1 patient had type 2 diabetes, but none of the primary Sjögren’s patients had a comorbid autoimmune disease. These data have been clarified in the revised manuscript (Lines 83-84 and Table 1).

Comment 2:Table 1. All study participants were women, likely due to the small sample size and gender bias of the disease. This may represent a limitation to the generalizability of the study, which must be duly mentioned in the discussion.

Response 2 We thank the reviewerfor his/her comment. We have included this limitation in the Discussion section of the revised manuscript (lines 372-375).

Findings from this study should be interpreted bearing in mind our study limitations which include the small population and the gender bias of the disease. In addition, our results should be confirmed by studies in larger populations with both female and male patients that mirror the general pSS patient population”.

Comment 3:Lines 283-285. At functional level the authors state that genes for butanoate metabolism were significantly depleted in pSS patients’ microbiota. How can this result be reconciled with the increased abundance of Clostridiaceae and other butyrate-producers (Figure 4)? Please add a comment in the discussion.

Response 3:Thank you for your remark. We have now added in the Discussion section a paragraph about why genes for butanoate metabolism were significantly depleted in the microbiota of pSS patients despite the increase in the abundance of Clostridiaceae and other butyrate-producing families in these study groups (Lines 349-356).

The depletion in genes involved in butanoate metabolism in the pSS patients could be due to the significantly lower abundance of the families Ruminococcaceae(in particular Faecalibacterium prausnitzii) and Lachnospiraceae(especially Roseburia spp.), the main butyrate producing-bacteria belonging to the phylum Firmicutes, despite the significantly higher abundance in these patients of other potential butyrate producers such as Clostridiaceae, Veillonellaceae and Prevotellaceae. Clostridiaceae, Prevotellaceae and Veillonellaceae comprise mainly bacteria that produce acetate, succinate and propionate, although some microorganisms of these families could promote butyrate production via acetate or succinate interconversion [43, 44]”. 

Comment 4:Lines 294-295 and 364-366. Was a vegan/vegetarian diet among the exclusion criteria of the study? This could especially be important in relation to the result relating to the Prevotella genus. An increased Prevotella in the intestine has been associated with the consumption of a plant- based diet (PMID: 26229982 and PMID: 25892234). This information should be added in the method and result sections or, if not available, added as a limitation in the discussion.

Response 4:Thank you for your pertinent comment.The vegetarian diet was not among the exclusion criteria of our study because all the pSS patients and healthy controls in our study followed the Mediterranean diet. Adherence to the Mediterranean diet was assessed via a validated 14-item food frequency questionnaire (Martinez-Gonzalez et al. PLoS One. 2012;7(8):e43134).This information has been added in the Method and Results sections of the revised manuscript (Lines 87-88).

“Adherence to the Mediterranean diet assessed via a 14-item food frequency questionnaire reflected high adherence to the Mediterranean diet in both study groups”.  

(Lines 388-392)

“All pSS patients and healthy controls included in the study followed a Mediterranean diet (rich in olive oil, nuts, fruits, vegetables, legumes, whole grains and fish, with a low intake of meat and dairy products). Adherence to the Mediterranean diet was assessed via a validated food frequency questionnaire composed of 14 questions regarding the main food groups consumed as part of the Mediterranean diet [51]”.

Comment 5:Lines 298-299. Experimental studies indicate that gut colonization by Prevotella does not in itself induce disruption to the intestinal barrier, but can lead to immune dysregulation when the barrier is damaged by other factors (see also PMID: 32433514). Hence, Prevotella could only be an enterotype marker without pathogenic role (PMID: 21508958).

Response 5:Thank you for your comment. Although Wright et al. demonstrated that Prevotellacontains enzymes that are important in mucin degradation, which may lead to increased intestinal permeability (Wright D. FEMS Microbiology Letters. 2000;190:73-79) (Lines 307-309), following the reviewer's comments we have added the following paragraph in the Discussion section of the revised manuscript (Lines 312-315).

Conversely, a recent study in immunocompetent mice has suggested that alteration of the gut microbiome by Prevotella spp. colonization decreases levels of short chain fatty acids, especially acetate, enhancing intestinal inflammation but after chemical damage to the intestinal barrier [33]. Accordingly, the significant increase in Prevotella abundance found in our study could be associated with the immune dysregulation and the increased intestinal inflammation in the pSS patients”.

Comment 6: Line 446. Statistical Analysis. The authors reported that the comparison between two groups was made with the Wilcoxon rank-sum test, and later, on lines 449-450, mention is made of the Mann-Whitney U test. Since the two names designate the same test, it is preferable to use only one wording. Please note that the Wilcoxon rank-sum test is not the Wilcoxon Signed Rank Test.

Response 6: As recommended by the reviewer, in the Statistical analysis section, we have  joined both paragraphs (line 446 and lines 449-450) to indicate  that the comparison between the two groups was made with the Mann-Whitney U test for continuous variables (Lines 463-464).

Minor points

Comment 7:Line 77. I suggest: “clinical characteristics”.

Response 7:The suggestion of the reviewer, “clinical characteristics”, has been included on line 78.

Comment 8:Table 1. Please indicate insulin concentration units.

Response 8:Insulin concentration units have been added to Table 1 in the revised manuscript.

Comment 9:Figure 1. In general, the scatter points are too large causing overlap between them. A slight reduction in size may be advisable.

Response 9: Unfortunately, it is not possible to changethe size of the scatter points in Figure1. But if the reviewer considers it necessary, we could eliminate all the scatter points from the plots.

Comment 10:Figure 3, 4 and 5. In the y axis, please correct “identificed” to “identified”.

Response 10:Titles on the y Axis of Figures 3, 4 and 5 have been corrected.

Comment 11:Figure 4. Please double check the names of the families in the x axis (eg. Bacteroidaceae).

Response 11:  The names of the families on the x axis of Figure 4 have been corrected.

Comment 12:Line 137. I suggest to replace “higher” with “more frequent”.

Response 12:“Higher” has been replaced with “more frequent” on lines 141-142.

Comment 13:Line 212. I suggest: “was significantly associated”.

Response 13: This has been changed to“was significantly associated” on line 212, now line 216.

Comment 14:Lines 355-357: These sentences are template instructions and must be deleted.

Response 14:The sentence on lines 355-357 has been deleted in the revised manuscript.

Comment 15:Line 440: Perhaps the authors meant 2−ΔΔCT method

Response 15:We thank the reviewer for this observation. This error has been corrected (Line 457).

Comment 16:Line 451: I suggest "linear regression analysis".

Response 16:  “Linear regression analysis” has been added on line 466.

References are not formatted correctly or include errors in the authors’ name. Please check all references accurately.

Comment:Line 511. Reference no. 1. The authors are: Mariette X, Criswell LA.

Response:We have corrected the authors of reference 1.

Comment:Line 512, line 514 and line 605. Please add the umlaut “ö” to the surname Sjögren.

Response:The umlaut “ö” has been added to the surname Sjögren in references 2, 3, 13 and 50 (line 528, 530, 555 and 637).

Comment:Line 517. The first author’s surname is Breban while Maxime is the name.

Response:The first author’s name in reference 2 has been corrected.

Comment:Line 521. The third author's name must have an acute accent on the “o”.

Response: Theacute accent on the “ó” of the third author's name in reference 6 has been added.

Comment:Line 537. Please add an umlaut “ä” to the surname of the seventh author.

Response:The umlaut “ä” has been added to the surname of the seventh author in reference 13.

Comment:Line 544. The surname of the third author is Ramirez Vinolo not just Vinolo. I also believe the last author's surname is “dos Santos Martin”.

Response: The name of the third author in reference 16 has been corrected.

Comment:Line 584. The abbreviated title of the journal is: Am J Physiol Gastrointest Liver Physiol.

Response:Reference 36 has been corrected.

Reviewer 2 Report

Gut microbiota has been reported to play a crucial role in many human diseases. Emerging evidence indicate that gut dysbiosis contributes to the pathophysiology or exacerbation of autoimmune diseases, including  Sjögren’s syndrome through the imbalance of the immune response.

During last decade many papers  have been released on gut  microbiome composition and its correlation with severity of disease and inflammatory markers  in Sjögren’s syndrome.

In presented papers authors aimed to investigate a new problem: connection between  intestinal dysbiosis, intestinal permeability and inflammatory markers and FOXP3 gene  expression in peripheral blood in  patients with Sjögren’s syndrome.

Overall the study is well done, succinctly written and nicely illustrated. The molecular methods are adequate and straightforward.

I have one point of concern that needs to be addressed, related to the authors' conclusions based on their results and the review of the literature.

Authors have found significant differences in predominant phyla, family, genera and species taxa in pSS patients compared to healthy controls, specially the significant increase in Prevotella abundance.  Authors concluded, that   it could be associated with the pathogenesis of primary Sjögren’s syndrome.  Interestingly, this difference is not consistent in other studies.  In study de Paiva at all  relative abundance of Bacteroides, Parabacteroides, Faecalibacterium, and Prevotella was reduced compared to controls. Because the gut microbiota composition is crucial for other results of the study and for final conclusions this discrepancy should by discussed in detail. I agree that nutrition is  the major factor that shapes the host’s gut microbiota but discrepancies cannot be explained only by differences in dietary habits ( the study de Paiva at all.  was conducted in USA). Certain changes in composition and diversity are associated with biological or functional age. There is a big range of age of SS patients (18-75 years) . Is there a dependency between age and microbiota differences?   How many SS patients have been treated hydroxychloroquine?

Minor comments:Rows 355-357 There is a sentence that do not fit to the text.  “Authors should discuss the results and how they can be interpreted in perspective of previous studies and of the working hypotheses. The findings and their implications should be discussed in the broadest context possible. Future research directions may also be highlighted”.

Author Response

Gut microbiota has been reported to play a crucial role in many human diseases. Emerging evidence indicate that gut dysbiosis contributes to the pathophysiology or exacerbation of autoimmune diseases, including Sjögren’s syndrome through the imbalance of the immune response.

During last decade many papers have been released on gut microbiome composition and its correlation with severity of disease and inflammatory markers in Sjögren’s syndrome.

In presented papers authors aimed to investigate a new problem: connection between intestinal dysbiosis, intestinal permeability and inflammatory markers and FOXP3 gene expression in peripheral blood in patients with Sjögren’s syndrome.

Overall the study is well done, succinctly written and nicely illustrated. The molecular methods are adequate and straightforward.

Response:

Comment 1:I have one point of concern that needs to be addressed, related to the authors' conclusions based on their results and the review of the literature.

Authors have found significant differences in predominant phyla, family, genera and species taxa in pSS patients compared to healthy controls, specially the significant increase in Prevotella abundance.  Authors concluded, that   it could be associated with the pathogenesis of primary Sjögren’s syndrome.  Interestingly, this difference is not consistent in other studies.  In study de Paiva at all relative abundance of Bacteroides, Parabacteroides, Faecalibacterium, and Prevotella was reduced compared to controls. Because the gut microbiota composition is crucial for other results of the study and for final conclusions this discrepancy should by discussed in detail. I agree that nutrition is  the major factor that shapes the host’s gut microbiota but discrepancies cannot be explained only by differences in dietary habits ( the study de Paiva at all.  was conducted in USA). Certain changes in composition and diversity are associated with biological or functional age. There is a big range of age of SS patients (18-75 years). Is there a dependency between age and microbiota differences?   How many SS patients have been treated hydroxychloroquine?

Response 1:First of all, we would like to thank the reviewer for his/her useful comments and suggestions, which undoubtedly have helped to improve our manuscript. As suggested, in this new version of the manuscript, we have added a paragraph relating to the reviewer's suggestions about the major factors responsible for the changes in the host’s gut microbiota between studies on Sjögren’s syndrome. Lines 297-301 in the revised manuscript.

 “The possible causes of these differences in gut microbiota composition between studies could be differences in dietary habits or in the severity of Sjögren’s syndrome, medications, age or sex bias. Several studies have described age- and sex-related changes in human gut microbiota composition [25-27]”.

Minor comments: Rows 355-357 There is a sentence that do not fit to the text.  “Authors should discuss the results and how they can be interpreted in perspective of previous studies and of the working hypotheses. The findings and their implications should be discussed in the broadest context possible. Future research directions may also be highlighted”.

Response:Thank you for your comment. The sentences on lines 355-357 have been eliminated in the Discussion section of the revised manuscript. In this revised version we have improved the discussion of the results in view of previous studies and of the working hypotheses.